taxonomy and systematics, evolution

biogeography, DNA barcode, neotropical, phylogenomic, Sphingidae

**Authors for correspondence:**
Xuankun Li
e-mail: xuankun.li@floridamuseum.ufl.edu
Akito Y. Kawahara
e-mail: kawahara@flmnh.ufl.edu

# A diversification relay race from Caribbean–Mesoamerica to the Andes: historical biogeography of *Xylophanes* hawkmoths

Xuankun Li[1], Chris A. Hamilton[1,3], Ryan St Laurent[1,4], Liliana Ballesteros-Mejia[5,6], Amanda Markee[1], Jean Haxaire[5], Rodolphe Rougerie[5], Ian J. Kitching[7] and Akito Y. Kawahara[1,2,8]

[1]McGuire Center for Lepidoptera and Biodiversity, Florida Museum of Natural History, and [2]Department of Biology, University of Florida, Gainesville, FL 32611, USA
[3]Department of Entomology, Plant Pathology and Nematology, University of Idaho, Moscow, ID 83844, USA
[4]Smithsonian National Museum of Natural History, Department of Entomology, Washington, DC 20560, USA
[5]Institut de Systématique, Evolution, Biodiversité (ISYEB), Muséum national d'Histoire naturelle, CNRS, Sorbonne Université, EPHE, Université des Antilles, Paris, France
[6]CESAB, Centre de Synthèse et d'Analyse sur la Biodiversité, Montpellier, France
[7]Department of Life Sciences, Natural History Museum, Cromwell Road, London SW7 5BD, UK
[8]Entomology and Nematology Department, University of Florida, Gainesville, FL 32608, USA

XL, 0000-0002-0622-2064; CAH, 0000-0001-7263-0755; RST, 0000-0001-6439-5249; LB-M, 0000-0003-2790-8652; JH, 0000-0001-6375-3892; RR, 0000-0003-0937-2815; IJK, 0000-0003-4738-5967; AYK, 0000-0002-3724-4610

The regions of the Andes and Caribbean-Mesoamerica are both hypothesized to be the cradle for many Neotropical lineages, but few studies have fully investigated the dynamics and interactions between Neotropical bioregions. The New World hawkmoth genus *Xylophanes* is the most taxonomically diverse genus in the Sphingidae, with the highest endemism and richness in the Andes and Caribbean-Mesoamerica. We integrated phylogenomic and DNA barcode data and generated the first time-calibrated tree for this genus, covering 93.8% of the species diversity. We used event-based likelihood ancestral area estimation and biogeographic stochastic mapping to examine the speciation and dispersal dynamics of *Xylophanes* across bioregions. We also used trait-dependent diversification models to compare speciation and extinction rates of lineages associated with different bioregions. Our results indicate that *Xylophanes* originated in Caribbean-Mesoamerica in the Late Miocene, and immediately diverged into five major clades. The current species diversity and distribution of *Xylophanes* can be explained by two consecutive phases. In the first phase, the highest *Xylophanes* speciation and emigration rates occurred in the Caribbean-Mesoamerica, and the highest immigration rates occurred in the Andes, whereas in the second phase the highest immigration rates were found in Amazonia, and the Andes had the highest speciation and emigration rates.

## 1. Introduction

The Neotropics is one of the most species-rich regions on Earth [1]. Biodiversity studies in the Neotropics have hypothesized both the Andes and Caribbean-Mesoamerica as cradle(s) for Neotropical lineages, but only a few studies have investigated the dynamics and interactions between bioregions [1,2]. The uplift of the Andes was a major event in the geological history of South America, providing barriers and opportunities for allopatric speciation and new ecological conditions for adaption and ecological speciation of animals (e.g. [2–4]) and flora (e.g. [5,6]). The Caribbean-Mesoamerican region has over 700 islands and a

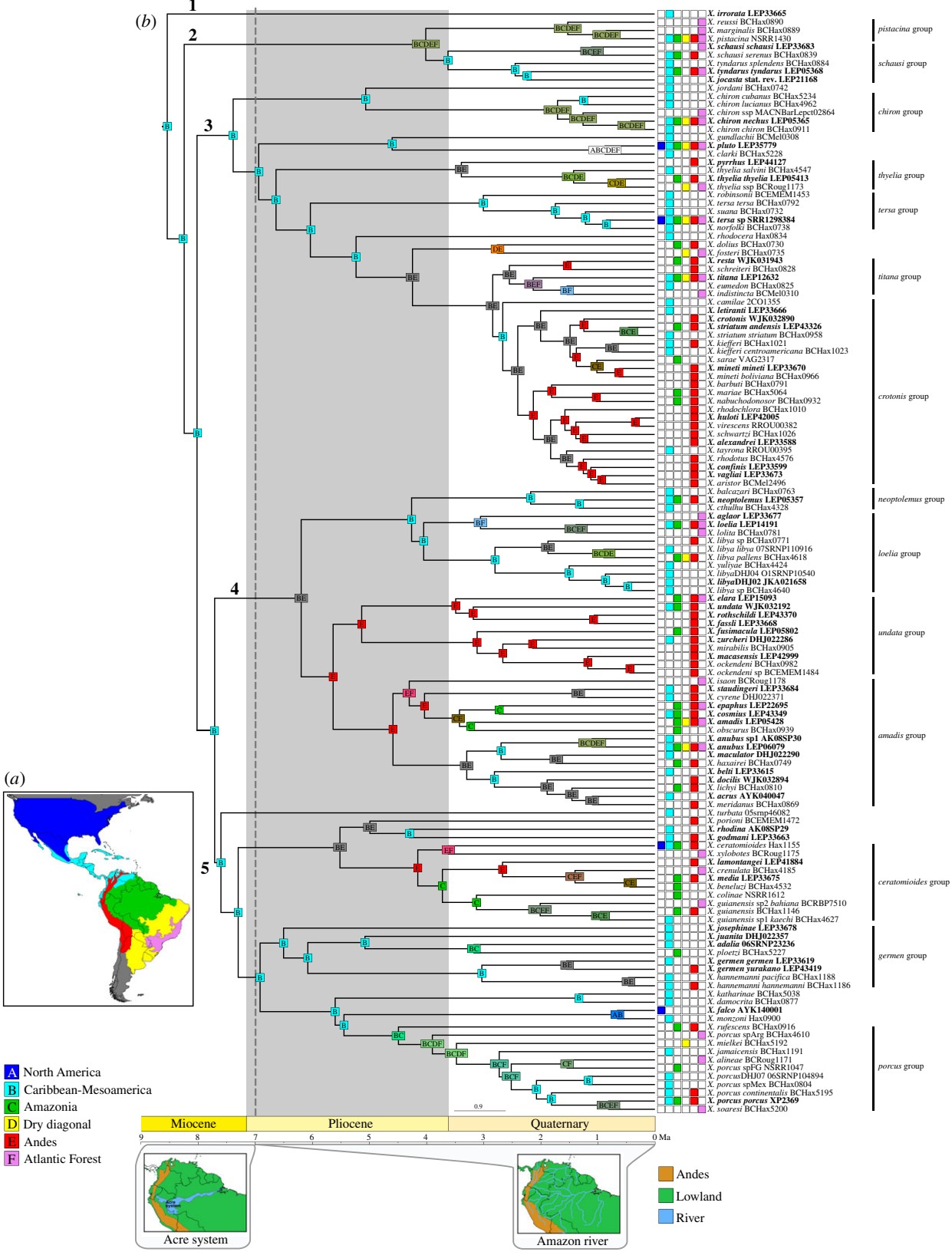

**Figure 1.** (*a*) Map of the Americas and the defined bioregions based on distribution data of *Xylophanes*. The delineated bioregions are mainly based on [48]. (*b*) Ancestral area estimates for *Xylophanes* under the dispersal–extinction–cladogenesis model and constrained dispersal rates (DEC, M1). The estimation was performed with BioGeoBEARS, based on the chronogram generated using BEAST shown in electronic supplementary material, appendix S4. Scale is in Ma. Distribution of each species is mapped to the right of the chronogram. A single most probable ancestral area is mapped at each node. Maps below the scale are modified from Hoorn *et al*. [49] showing palaeogeographical models of two time slices used in the constrained analysis. Left: 7–11 Ma, Panama Isthmus open, Acre system present and northern Andes undeveloped; right: 7 Ma to the present, Panama Isthmus closed and northern Andes developed. (Online version in colour.)

land bridge connecting two major continents [7]. Studies have identified this region as important for *in situ* speciation (e.g. [8,9]) with lineages originating in Caribbean-Mesoamerica and dispersing to South America (e.g. [7,10–12]). There are five biogeographic processes that describe Neotropical biodiversity: *cradle* (high speciation rate), *museum* (low extinction

**Table 1.** Dispersal rates matrix between each pair of biogeographic areas considered and for the two time slices used in our historical biogeography analysis. Basic dispersal rates: 0.5 = regions are contiguous; 0.1 = two regions separated by another; 0.01 = two regions separated by more than two regions. Additional low dispersal rates: 0.1 = two regions separated by water; 0.01 = two regions separated by water and other regions. Additional high dispersal rates: 0.7 = between undeveloped nothern Andes and mesoamerican low lands (11–7 Ma).

| | manual_dispersal_multipliers | | | | | |
| | A | B | C | D | E | F |
|---|---|---|---|---|---|---|
| 7 Ma to present | | | | | | |
| A | 1 | 0.5 | 0.1 | 0.01 | 0.1 | 0.01 |
| B | 0.5 | 1 | 0.5 | 0.1 | 0.5 | 0.01 |
| C | 0.1 | 0.5 | 1 | 0.5 | 0.5 | 0.1 |
| D | 0.01 | 0.1 | 0.5 | 1 | 0.5 | 0.5 |
| E | 0.1 | 0.5 | 0.5 | 0.5 | 1 | 0.1 |
| F | 0.01 | 0.01 | 0.1 | 0.5 | 0.1 | 1 |
| 11–7 Ma | | | | | | |
| A | 1 | 0.5 | 0.01 | 0.01 | 0.01 | 0.01 |
| B | 0.5 | 1 | 0.5 | 0.1 | 0.7 | 0.01 |
| C | 0.01 | 0.5 | 1 | 0.1 | 0.5 | 0.01 |
| D | 0.01 | 0.1 | 0.1 | 1 | 0.5 | 0.5 |
| E | 0.01 | 0.7 | 0.5 | 0.5 | 1 | 0.1 |
| F | 0.01 | 0.01 | 0.01 | 0.5 | 0.1 | 1 |

rate), *time-for-speciation* (early colonization), *sink* (as 'species-attractor', high immigration rate) [13] and *source* (high emigration rate) [1]. These processes play different but not necessarily mutually exclusive roles in shaping current biodiversity patterns.

With more than 120 described species, *Xylophanes* is the most species-rich genus of hawkmoths [14,15]. The taxonomy has been carefully revised with the combination of morphology and DNA barcode data (e.g. [16,17]). These moths are pollinators and strong fliers and are thought to have high dispersal ability [18]. *Xylophanes* belongs to the mostly Old World subtribe Choerocampina [19,20], suggesting that *Xylophanes* may have dispersed to the New World via a jump dispersal event [19], or that the Choerocampina was already widely distributed globally prior to the origin of *Xylophanes* and the majority of the New World representatives then went extinct during the Neogene cooling [21].

*Xylophanes* is an ideal candidate to study dynamics and interactions between Neotropical bioregions because of the extraordinarily high species-level diversity for a hawkmoth lineage, the wide distribution of the genus covering all Neotropical bioregions, the relatively restricted geographical ranges of some individual species, and the excellent availability of DNA barcodes. We integrated phylogenomic and DNA barcode data to generate the first time-calibrated tree for *Xylophanes*, covering 93.8% of the species diversity. We used trait-dependent diversification models to compare speciation and extinction rates of lineages associated with different bioregions. We also used event-based likelihood ancestral area estimation, and biogeographic stochastic mapping to examine the speciation and dispersal dynamics among bioregions to provide insights into the evolutionary mechanisms underlying the diversification of *Xylophanes* in the Neotropics.

## 2. Material and methods

### (a) Taxon sampling

In the present study, 150 taxa were selected for phylogenetic analysis, including (i) 136 operational taxonomic units (OTUs), comprising both described species and subspecies, and the Barcode Index Numbers [22] that we consider as representing currently unrecognized (sub)species of *Xylophanes*; and (ii) 14 non-*Xylophanes* outgroups (electronic supplementary material, appendix S1). Anchored Hybrid Enrichment (AHE) data were newly generated for 57 taxa (53 ingroup and four outgroup) using the BOM1 Agilent Custom SureSelect probe set [23]. Cytochrome c oxidase subunit I (CO1) barcode data of all ingroup taxa were added to our phylogeny, including 28 sequences generated here and 109 generated as part of a global DNA barcoding campaign for Sphingidae (electronic supplementary material, appendix S1); all DNA barcodes are publicly available from BOLD dataset DS-XYLOPHY1 (doi:10.5883/DS-XYLOPHY1). Specimens were identified by co-authors A.Y.K., I.J.K., J.H. and R.R. using morphology and CO1 barcodes. Ingroups sampled represent 136 of the 145 known *Xylophanes* OTUs and 93.8% of species diversity. Details of the sampled taxa can be found in electronic supplementary material, appendices S1 and S2.

### (b) DNA extraction and the BOM1 anchored hybrid enrichment probe set

DNA extractions for 63 taxa were conducted for the present study. These extractions were used for AHE sequencing, following the methods outlined in [23], and targeting 571 loci across Bombycoidea, including the CO1 barcode. AHE, library preparation, hybrid enrichment, and sequencing were carried out at RAPiD Genomics (Gainesville, FL, USA). CO1 barcodes of seven samples were amplified by PCR using the LCO/HCO universal insect primers [24] and NEB Long Taq DNA polymerase (New England BioLabs, Ipswich, Suffolk, UK) at the McGuire

Proc. R. Soc. B 289: 20212435

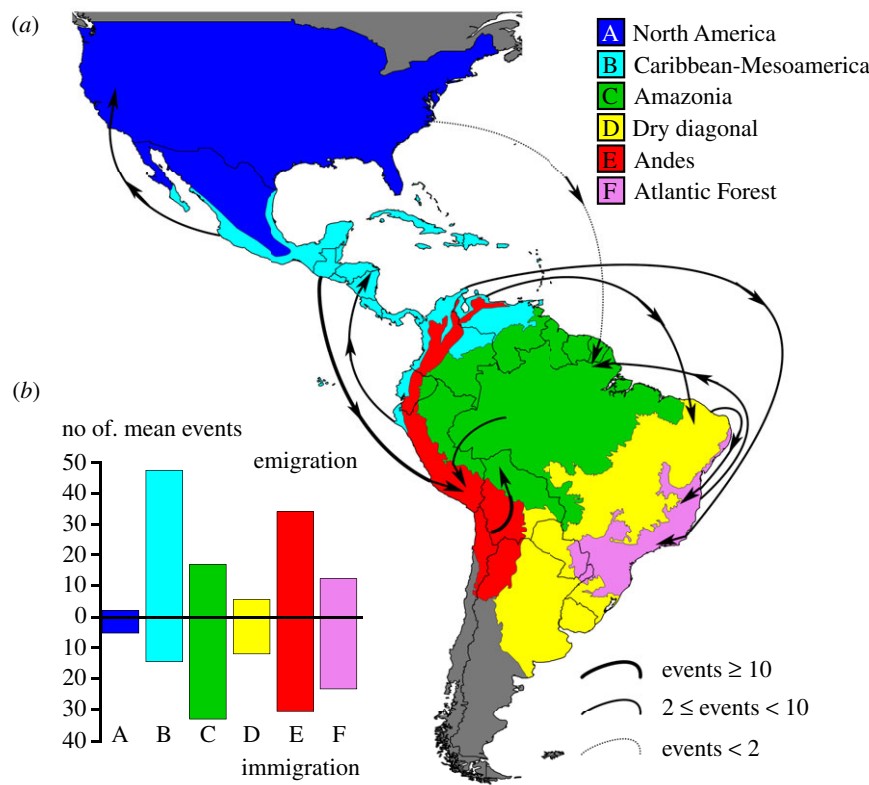

**Figure 2.** Summary of major *Xylophanes* dispersal events, average number of dispersal events between two areas based on 100 000 biogeographic stochastic mappings under the DEC model in BioGeoBEARS. (*a*) The highest emigration and the highest immigration of each area are summarized on the map. The width and shape of lines represent the estimated average number of dispersal events. (*b*) Bar chart showing average number of emigration and immigration events of each area. Complete average dispersal events between each type shown in electronic supplementary material, appendix S5. (Online version in colour.)

**Table 2.** Marginal-likelihood estimate (MLE) scores for various BEAST analyses performed for this study, and estimated ages (in Ma) for *Xylophanes* crown nodes for each tree prior/clock scheme in BEAST. Notes: SS, stepping-stone sampling marginal-likelihood estimation; PS, path-sampling marginal-likelihood estimation; median post-burn-in divergence times in millions of years (95% credibility interval).

| analysis | tree model | clock model | MLE SS | MLE PS | crown *Xylophanes* age |
|---|---|---|---|---|---|
| A1 | birth–death | 1 ULRC | −157 592.8747 | −157593.5007 | 13.1023 (10.8207, 15.0035) |
| A2 | Yule | 1 ULRC | −157589.6111 | −157589.9523 | 13.1493 (10.9506, 14.9813) |
| A3 | birth–death | 3 ULRC | −150523.204 | −150523.6057 | 11.4704 (9.4005–13.6747) |
| A4 | Yule | 3 ULRC | −150519.484 | −150519.9143 | 11.6692 (9.6526–13.7525) |
| A5 | birth–death | 11 ULRC | −149066.5954 | −149066.1731 | 8.5302 (7.8203–9.4886) |
| A6 | Yule | 11 ULRC | −149074.3488 | −149074.0018 | 8.9603 (7.9862–10.0528) |

Center for Lepidoptera and Biodiversity, Florida Museum of Natural History (MGCL) (electronic supplementary material, Appendix S1), and sequenced at Eurofins Genomics (Louisville, KY, USA). All DNA extracts are stored at −80°C in the molecular collection of the MGCL, in Gainesville, FL, USA.

## (c) Dataset preparation

For BOM1 'probe' regions with phylogenomic data (68 species: 10 transcriptomes and 58 AHE sequences), we followed the assembly steps outlined in [25]. Raw reads were assembled with Trim Galore! v.0.4.0 (bioinformatics.babraham.ac.uk). Orthology was determined using the *Bombyx mori* genome [26] as reference using with NCBI blastn [27]. Cross-contamination checks were conducted with USEARCH [28]. Cleaned sequences were aligned in MAFFT v. 7.245 [29], and isoform consensus sequences were generated using FASconCAT-G 1.02 [30]. We used a long-branch detection protocol to investigate possible non-orthologous sequences followed [31] (for details, see electronic supplementary material, appendix S2). Individual locus information was summarized using AMAS [32] and loci with less than 60% taxon coverage (41 taxa) were excluded. In total, 482 loci were selected across 68 taxa.

New DNA barcode sequences were either captured with the AHE probe set (in 21 taxa), or Sanger sequenced (in 7 taxa); their identification was verified using BOLD Identification Engine to rule out contamination issues or misidentification. All these DNA barcode sequences were checked for 5′ to 3′ direction, then aligned in MAFFT v. 7.245 [29] together with the 109 sequences downloaded from BOLD. All nucleotide sequences were then manually examined in AliView [33] to ensure correct reading frame and corresponding amino acid alignments. Cleaned MSAs of each locus were concatenated using Phyx v. 1.1 [34] to generate a matrix with 150 taxa and 483 loci (122 100 aligned nucleotides).

## (d) Phylogenetic analysis

Phylogenetic analyses using an ML approach with 60 separate heuristic searches were carried out in IQ-TREE v. 2.1.2 [35]. The matrix was partitioned by locus, and the best partitioning scheme determined by allowing merging of partitions ('-MFP + MERGE' command) and using the Bayesian Information Criterion (BIC). Details on parameter settings, see electronic supplementary material, appendix S2. Node supports were computed via 1000 ultrafast bootstrap ('-B 1000' command) replicates [36,37], and SH-aLRT ('-alrt 1000' command) [38]. We refer to support as 'strong' if SH-aLRT ≥ 80 and UFBoot ≥ 95, and 'moderate' if SH-aLRT ≥ 80 or UFBoot ≥ 95, following [36].

## (e) Divergence time estimation

Divergence time estimation was implemented in a Bayesian framework using BEAST v. 1.10.4 [39]. We used SortaDate [40] to reduce the nucleotide alignment to a computationally tractable matrix (50 loci), and used PartitionFinder2 [41] to partition the reduced matrix (details see electronic supplementary material, appendix S2). The reduced concatenated data matrix was imported into BEAUTi (BEAST package). Substitution models were unlinked among partitions, and clock and tree models were linked. We applied both an uncorrelated relaxed molecular clock model [42] and an exponential prior. We also tested two different tree priors, Yule (pure birth) and birth–death for each partition schemes. We used a fixed cladogram based on the best topology selected from the previous IQ-TREE analyses. Nine nodes were constrained with uniform distributions based on the 95% confidence interval (CI) given in [43]. Details on calibration nodes selected are provided in electronic supplementary material, appendix S3.

Three independent runs of each clock scheme and tree prior combination were run to check for convergence. The subsampled trees were used to summarize the maximum clade credibility tree by TreeAnnotator [44], with median heights as node heights. In order to identify the best tree prior and clock scheme combination, path sampling and stepping stone sampling [45–47] were performed as part of all BEAST analyses. Details on parameter settings see electronic supplementary material, appendix S2.

## (f) Ancestral area estimation

Distribution areas for each *Xylophanes* species (see electronic supplementary material, appendix S1) for the BioGeoBEARS analyses were assessed by I.J.K. and R.R. (see electronic supplementary material, appendix S2). We recognize six bioregions that best account for the distribution of the species within the genus: (A) North America, (B) Caribbean-Mesoamerica, (C) Amazonia, (D) Dry diagonal, (E) Andes and (F) Atlantic Forest (figure 1a), which largely follows widely accepted scheme of [48].

We performed an event-based likelihood ancestral area estimation using BioGeoBEARS [50]. Three models were used: (1) DEC (Dispersal Extinction Cladogenesis; [51]); (2) DIVALIKE (a likelihood-based implementation of dispersal vicariance analysis, originally parsimony based; [52]); and (3) BAYAREALIKE (a likelihood implementation of BayArea, originally Bayesian; [53]). All models were also evaluated under a constrained analysis (M1), in which we considered palaeogeographical events that occurred in the past 11 Myr over two time slices (11–7 Ma and 7 Ma to present; formation of the Panama Isthmus, Acre system and the orogeny of the Andes) and geographical distance variation (table 1b and figure 2), for a total of six scenarios. The Akaike Information Criterion (AIC, [54]) and the corrected Akaike Information Criterion (AICc, [55]) were calculated. The chronogram from BEAST was used for this analysis after exclusion of outgroup taxa and of two *Xylophanes* OTUs lacking distribution data (*X. hannemanni* sp1 and *X. hannemanni* sp2; electronic supplementary material, appendix S1), thus leaving 134 ingroup taxa.

**Table 3.** Results of the BioGeoBEARS analyses.

| model | palaeogeographical constraint | numparams | LnL | d | e | j | AIC | ΔAIC | AIC_wt | AICc | ΔAICc | AICc_wt |
|---|---|---|---|---|---|---|---|---|---|---|---|---|
| DEC | unconstrained | 2 | −387.46 | 0.05 | 0.00 | 0.00 | 778.93 | 7.48 | $2.32 \times 10^{-2}$ | 779.02 | 7.48 | $2.32 \times 10^{-2}$ |
| DIVALIKE | unconstrained | 2 | −509.99 | 0.01 | 0.01 | 0.00 | 1023.98 | 252.54 | $1.42 \times 10^{-55}$ | 1024.07 | 252.54 | $1.42 \times 10^{-55}$ |
| BAYAREALIKE | unconstrained | 2 | −441.26 | 0.04 | 0.25 | 0.00 | 886.52 | 115.07 | $1.00 \times 10^{-25}$ | 886.61 | 115.07 | $1.00 \times 10^{-25}$ |
| DEC | constrained | 2 | −383.72 | 0.16 | 0.00 | 0.00 | 771.44 | 0.00 | $9.77 \times 10^{-1}$ | 771.54 | 0.00 | $9.77 \times 10^{-1}$ |
| DIVALIKE | constrained | 2 | −397.05 | 0.20 | 0.00 | 0.00 | 798.10 | 26.66 | $1.59 \times 10^{-6}$ | 798.20 | 26.66 | $1.59 \times 10^{-6}$ |
| BAYAREALIKE | constrained | 2 | −434.36 | 0.14 | 0.27 | 0.00 | 872.71 | 101.27 | $9.99 \times 10^{-23}$ | 872.81 | 101.27 | $9.99 \times 10^{-23}$ |

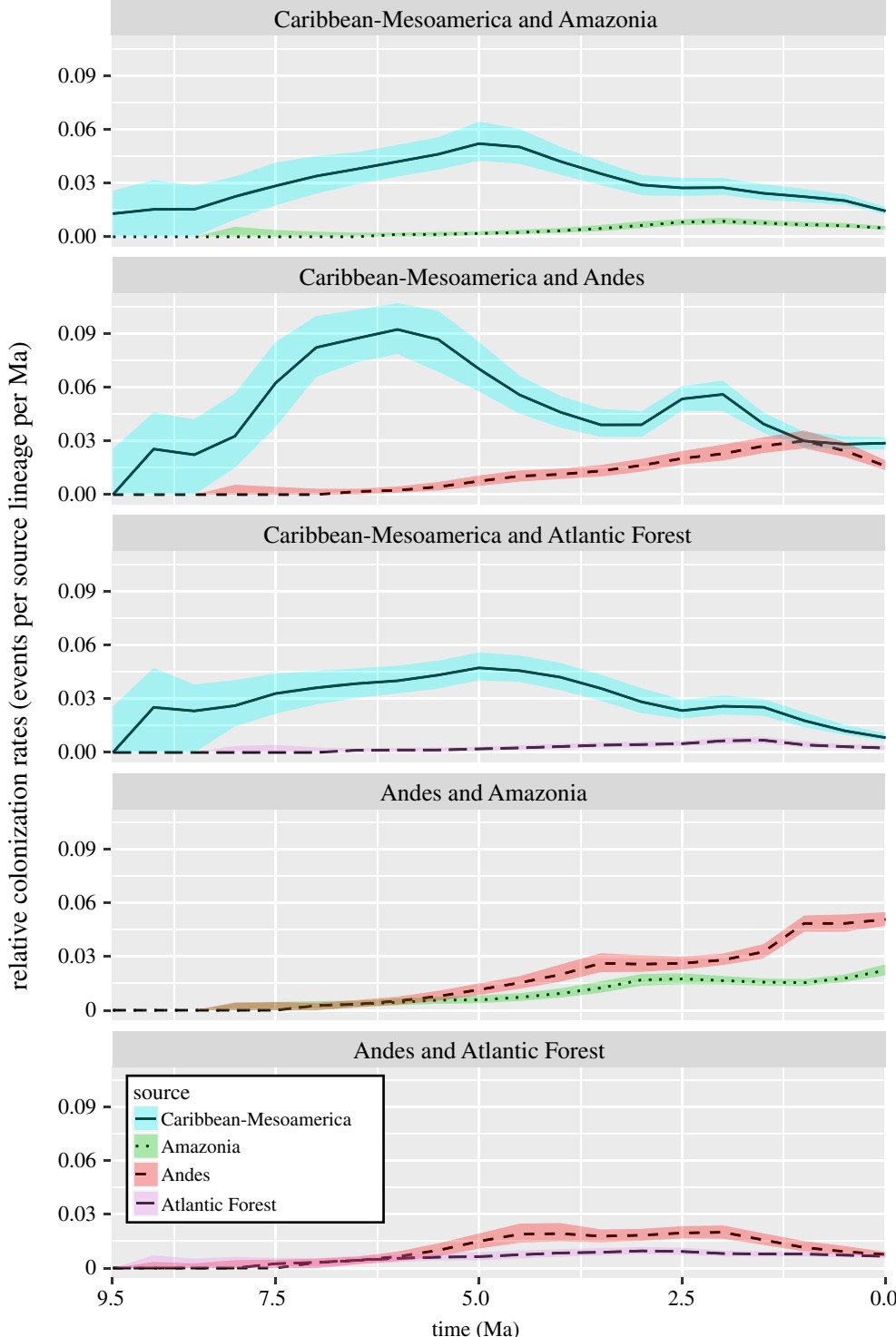

**Figure 3.** Dispersal rates through time based on 100 000 biogeographic stochastic mappings under the DEC model in BioGeoBEARS. Rates are displayed for selected pairs of areas. Source lines are the median values; coloured ribbons are the lower and upper quartiles (0.25 and 0.75 quantiles). (Online version in colour.)

## (g) Dispersal and speciation rates through time

To account for missing taxa and uncertainties related to divergence time estimation and ancestral areas we used simulated trees and carried out 100 biogeographic stochastic mappings (BSMs; [56]) for each of the new trees. In all, 100 000 pseudoreplicated biogeographic histories were simulated to estimate the number of dispersal events and *in situ* speciation events (details see electronic supplementary material, appendix S2). We used the DEC model implemented in BioGeoBEARS [50] to infer geographical range evolution of lineages and performed the analysis without any constraints to decrease artificial influence. We followed [57] to calculate *in situ* speciation rates as $\lambda_X(t1) = s_X(t1)/L_X(t0)$. We followed [1] to calculate the colonization rates as

$c_{XtoY}(t1) = d_{XtoY}(t1)/Br(t1)$. In addition, we calculated emigration rates as $E_X(t1) = df_X(t1)/Br(t1)$ and immigration rates as $I_X(t1) = dt_X(t1)/Br(t1)$ (details see electronic supplementary material, appendix S2).

## (h) State-dependent speciation and extinction

We applied 33 GeoHiSSE models [58] in the R package HiSSE [59] to study the effects of distribution on *Xylophanes* diversity. Four regions with high *Xylophanes* diversity (Amazonia, Andes, Atlantic Forest and Caribbean-Mesoamerica) were tested. Thirty-three models (adopted and modified from [58]; electronic supplementary material, appendix S4) were fitted, with the pruned chronogram and distribution characters modified from

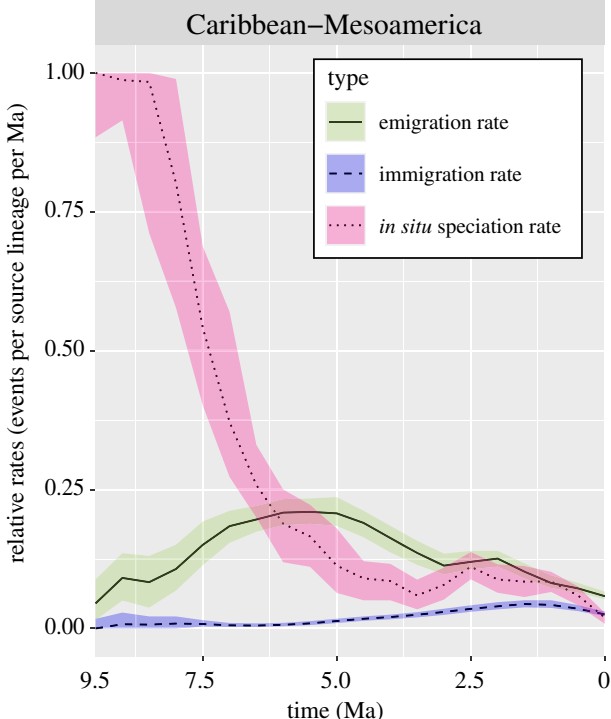

**Figure 4.** Within-area dispersal and speciation rates through time of Caribbean-Mesoamerica based on 100 000 biogeographic stochastic mappings under the DEC model in BioGeoBEARS. Type lines are the median values; coloured ribbons are the lower and upper quartiles (0.25 and 0.75 quantiles). (Online version in colour.)

the ancestral area estimation analyses (electronic supplementary material, appendix S1). Species were coded as endemic to the target region (state 1) or absent from the target region (state 2) or distributed in the target region and other regions (state 0). Details on sampling fractions for each bioregion, see electronic supplementary material, appendix S2. Finally, model-averaged diversification rates were mapped based on Akaike weights [58]. All phylogenetic and biogeographic analyses were conducted on the University of Florida HiPerGator High Performance Computing Cluster (http://www.hpc.ufl.edu/).

## 3. Results

The best partitioning schemes combined loci into 24 partitions (electronic supplementary material, appendix S5). *Xylophanes* was recovered as monophyletic (SH-aLRT/UFBoot: 100/99) and five major subclades were identified (figure 1b). Support for 36.6% and 38.6% of nodes were strong or moderate respectively. Best partitioning schemes for three different initial partition strategies for the BEAST analyses are listed in electronic supplementary material, appendix S5. The preferred BEAST analysis was identified with marginal-likelihood estimations (table 2), which supported a birth–death tree prior with 11 unlinked molecular clocks. Divergence time estimation results reveal an origin of *Xylophanes* in the Late Miocene at 8.6 Ma (95% highest posterior density = 7.8–9.5 Ma), immediately followed by the divergence of the five major clades around 7.7 Ma (7.0–8.6 Ma) (electronic supplementary material, appendix S6).

The biogeographic model DEC, with the M1 constrained analysis, yielded the highest likelihood among all six models tested (table 3 and figure 1b). This model recovered a Caribbean-Mesoamerica origin for the genus and for three of the five major clades, and Caribbean-Mesoamerica had the longest time for speciation (supporting the time-for-speciation hypothesis).

The simulation analysis recovered a pattern that is strongly consistent with the results of the ancestral area estimation analysis (figure 2; electronic supplementary material, appendix S7). Caribbean-Mesoamerica is the largest source for species dispersal to all other areas, followed by the Andes and Amazonia. Dispersal to the Andes from Caribbean-Mesoamerica occurred on average 18.3 times, the highest of all dispersal types. Amazonia is the largest sink followed by the Andes and the Atlantic Forest, and the Andes is the largest source for Amazonia with an average of 15.5 dispersal events (figure 2; electronic supplementary material, appendix S7).

Two consecutive phases are identified from the results. In the first phase (9.5 to approx. 2 Ma), dispersal from Caribbean-Mesoamerica was the highest, and the Andes was the greatest sink (figures 3 and 4). The *in situ* speciation rate in Caribbean-Mesoamerica was also the highest among all areas, although the rate experienced a sharp decrease until 5 Ma (figure 4). During the second phase (approx. 2 Ma to present), emigration and *in situ* speciation rates of the Andes exceeded Caribbean-Mesoamerica and became the highest (figures 3 and 5) while Amazonia simultaneously became the greatest sink (figure 3). Diversification and immigration/emigration patterns of *Xylophanes* followed a relay race-like pattern such that these important evolutionary dynamics shifted geospatially and temporally. A relay race of diversification and emigration was from Caribbean-Mesoamerica to the Andes, and a relay race of immigration was from the Andes to Amazonia.

GeoHiSSE models with the highest likelihood for Caribbean-Mesoamerica and the Andes had two rate classes, with or without extinction (electronic supplementary material, appendix S8). Lineages endemic to Caribbean-Mesoamerica had low relative speciation rate (0.13) and a low extinction rate ($2.51 \times 10^{-8}$), resulting in a low net-diversification rate (0.13) (electronic supplementary material, appendix S9). By contrast, lineages endemic to the Andes had a high relative speciation rate (0.56) and a high relative extinction rate (0.06), resulting in a relatively high net-diversification rate (0.5) (electronic supplementary material, appendix S9). Slightly lower net-diversification rates are found in endemic Amazonia and Atlantic Forest lineages (electronic supplementary material, appendix S10).

## 4. Discussion

Our study is the first comprehensive phylogenetic analysis of *Xylophanes* hawkmoths and presents the most likely biogeographic scenario for their diversification. We recovered all 14 previously recognized species groups as monophyletic, among which 12 were strongly or moderately supported (figure 1b; electronic supplementary material appendix S6). In trait-dependent diversification models, constrained analyses (M1) always yielded higher AICc weights than unconstrained analyses (M0) (table 3), indicating that the dispersal was not hampered by the still open Panama Isthmus but facilitated by the developing northern Andes, while the presence of the Acre system decreased the dispersal rate. Our historical biogeography reconstructions reveal that *Xylophanes* originated in Caribbean-Mesoamerica in the Late Miocene, setting the stage for a relay race of temporally and spatially shifting diversification patterns that occurred over the next 8 Myr. The 'baton' of high diversification and emigration rates in Caribbean-Mesoamerica and the Andes which led to successively high immigration rates in the

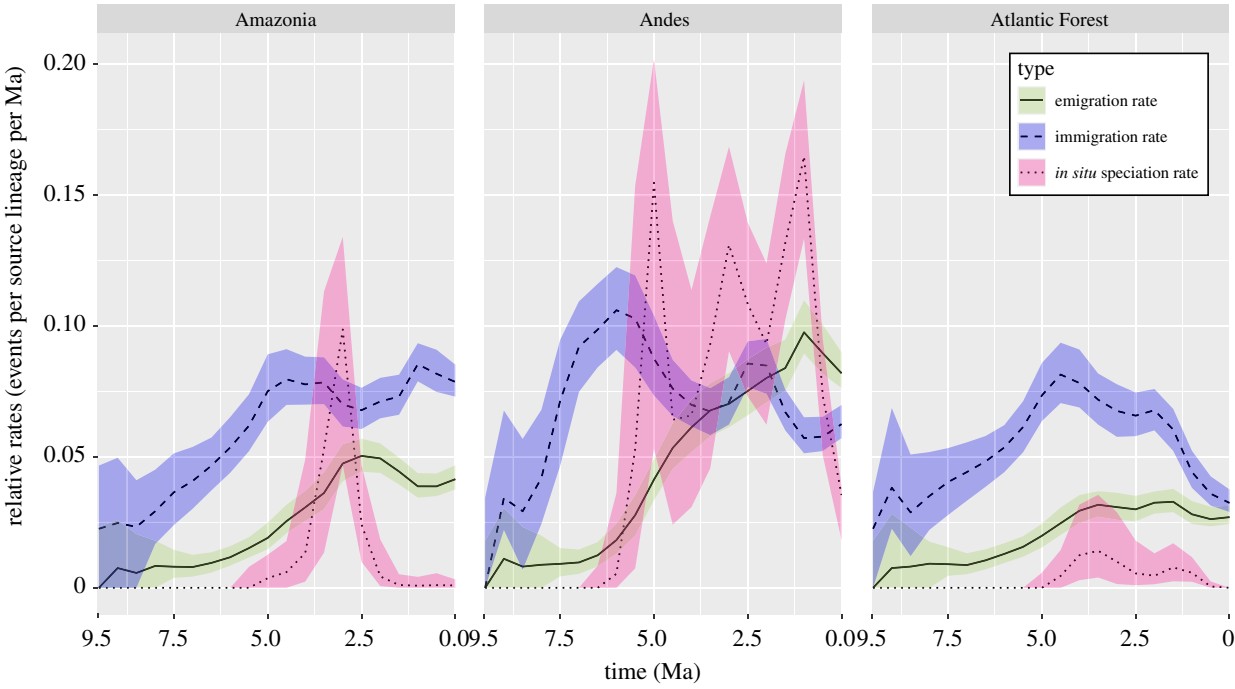

**Figure 5.** Within-area dispersal and speciation rates through time of Amazonia, Andes and Atlantic Forest based on 100 000 biogeographic stochastic mappings under the DEC model in BioGeoBEARS. Type lines are the median values; coloured ribbons are the lower and upper quartiles (0.25 and 0.75 quantiles). (Online version in colour.)

Andes and Amazonia, helps to explain the high diversity of *Xylophanes* in each of these major biogeographic regions of the Tropical Americas. Overall, we show that the high diversity of *Xylophanes* in tropical regions does not fall into a single, simple category of cradle, museum, time-for-speciation, sink or source over time, but that these vagile moths, which evolved during complex geological processes, likewise diversified and dispersed dynamically.

A Caribbean-Mesoamerica origin is unusual among Neotropical Lepidoptera, as previous research generally recover origins in either the historically stable Amazonia or the dynamic orogeny of the Andes (e.g. [2,60] but see [10,61,62]). Our results show that Caribbean-Mesoamerica is the largest source (figure 2; electronic supplementary material, appendix S7), and the cradle (during the first phase) for *Xylophanes* (figure 4). Similar scenarios have also been found in other Neotropical insect groups (e.g. [9,11,61]). During a second phase, a low speciation rate is found in Caribbean-Mesoamerica lineages (figure 4), and our GeoHiSSE tests support the 'museum' hypothesis of Caribbean-Mesoamerica for *Xylophanes* (electronic supplementary material, appendix S9). By contrast, a study of the Neotropical butterfly tribe Brassolini [2] found a high and increasing speciation rate in Caribbean-Mesoamerica lineages. This difference between two lepidopteran groups maybe due to Caribbean-Mesoamerica being an unstable environment for Brassolini, but a stable environment for *Xylophanes*. Because *Xylophanes* is much younger than Brassolini (8.6 versus 38 Ma) and experienced less geographical dynamism within Caribbean-Mesoamerican, *Xylophanes* are thought to be more vagile than most butterflies [18], which resulted in fewer instances of isolation for ancient *Xylophanes* in Caribbean-Mesoamerica compared to brassoline butterflies. For *Xylophanes*, Caribbean-Mesoamerica was a stable environment during the second phase, therefore, low speciation and extinction rates are expected [63,64]. Our results support the idea that an area considered variable for some species might be seen as stable for others [64]. The dynamics of dispersal rate from Caribbean-Mesoamerica to the Andes is coincident

with the intense orogeny of the Andes during the Miocene–Pliocene and Early Quaternary, when new ecological niches arose and facilitated colonization and diversification [13,49].

State-dependent speciation and extinction analyses identified the Andes as a cradle of *Xylophanes* diversity, but differences in speciation/extinction rates were driven by an unmeasured, hidden state. This result is unsurprising as ecological speciation has been shown to play an important role in insect diversification (e.g. [62,65]). The Andes is the second-largest source of *Xylophanes*, with most immigrations taking place from Caribbean-Mesoamerica during the Pliocene (figure 2; electronic supplementary material, appendix S7). Similar high immigration rates in Andean lineages have been found in other study systems, such as in birds and butterflies [60,66,67]. The *in situ* speciation rate of the Andes also surpassed Caribbean-Mesoamerica in the Mid-to-Late Quaternary (figures 4 and 5). The Andes orogeny has been shown to increase both the immigration rate and the *in situ* speciation rate of other lineages [13,49]. As the source in Mid-to-Late Quaternary, most of the emigrations from the Andes dispersed to Amazonia (figures 2 and 3; electronic supplementary material, appendix S7). Our study indicates that Amazonia is the largest sink and its high diversity was likely driven by immigration rather than *in situ* speciation. Amazonia as the largest sink has not been formally reported so far, although several studies have detected dispersal from the Andes to Amazonia (e.g. [49,60,66,67]). Our study offers a new perspective of Lepidoptera evolution in the Americas, where an incredibly diverse, widespread lineage of moths has undergone discontinuous, yet connected, periods of evolutionary dynamism in geographically separate regions with distinct topographic histories. The complex pattern of *in situ* diversification, with bouts of significant emigration and immigration events to colonize vast new areas of concurrently evolving landscapes, paints a vibrant picture of relatively recent events that shaped the largest radiation of hawkmoths on the planet. Furthermore, our results provide the foundation to understand the Neotropical component of the evolution of the mostly Old World

distributed Choerocampina, paving the way for future research that could uncover evolutionary parallels in the Old World by studying moths ecologically similar to *Xylophanes*, but which underwent wholly different biogeographic dynamics.

Data accessibility. The data and metadata associated with this article are available from the Dryad Digital Repository: https://doi.org/10.5061/dryad.mw6m905xp [68].

Authors' contributions. X.L.: data curation, formal analysis, methodology, visualization, writing—original draft, writing—review and editing; C.A.H.: conceptualization, resources, writing—review and editing; R.S.L.: methodology, software, writing—review and editing; L.B.-M.: data curation, formal analysis, software, writing—review and editing; A.M.: data curation, writing—review and editing; J.H.: validation, writing—review and editing; R.R.: data curation, formal analysis, funding acquisition, writing—review and editing; I.J.K.: data curation, funding acquisition, writing—review and editing;

A.Y.K.: conceptualization, funding acquisition, investigation, project administration, resources, supervision, writing—review and editing.

All authors gave final approval for publication and agreed to be held accountable for the work performed therein.

Competing interests. We declare we have no competing interests.

Funding. National Science Foundation (NSF) grant no. IOS #1920895 to A.Y.K. French National Research Agency (ANR) SPHINX grant no. (ANR-16-CE02–0011-01) to R.R. French Foundation for Research on Biodiversity (FRB) and CESAB synthesis centre to ACTIAS project (R.R., L.B.-M. and I.J.K.).

Acknowledgements. David Plotkin helped with the GeoHiSSE analyses. Jesse Barber and his lab at Boise State University helped sample some of the specimens in this study. Yixin Li (Canberra), Kok Ben Toh and Xinyuan Yang helped with R scripts. Thanks to contributors to the Global DNA barcoding campaign for Sphingidae, and support by the Canadian Centre for DNA Barcoding and Centre for Biodiversity Genomics at University of Guelph (Ontario, Canada) through the iBOL project.

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
