## [Peer Review File · Proceedings of the Royal Society B: Biological Sciences]

Review History

RSPB-2021-2435.R0 (Original submission)

Review form: Reviewer 1

Recommendation

Accept with minor revision (please list in comments)

Scientific importance: Is the manuscript an original and important contribution to its field?

Good

General interest: Is the paper of sufficient general interest?

Good

Quality of the paper: Is the overall quality of the paper suitable?

Excellent

Is the length of the paper justified?

Yes

Should the paper be seen by a specialist statistical reviewer?

Yes

Do you have any concerns about statistical analyses in this paper? If so, please specify them explicitly in your report.

No

It is a condition of publication that authors make their supporting data, code and materials available - either as supplementary material or hosted in an external repository. Please rate, if applicable, the supporting data on the following criteria.

Is it accessible?

No

Is it clear?

N/A

Is it adequate?

N/A

Do you have any ethical concerns with this paper?

Yes

Comments to the Author

I like this paper and recommend its publication in Proceedings B. Both the genomic and taxonomic sampling are comprehensive and enable assessing the addressed questions well. I also did not detect any flaws with the conducted analyses although I am not familiar with all of them. The pictures provided are nice and informative. My only comment is that I find that the paper would benefit for putting the problem into a wider context. The scope of this paper is a bit narrow presently. Why is it important to understand phylogeography of these hawkmoths? Certainly this is of interest because it provides important insights on the species diversification more generally in the Neotropical and Mesoamerican areas. I think that the big question of biogeography is that why is there disproportionately high number of species in the Neotropical area compared to other areas, including tropical Asia and Africa. I find that expanding the introduction, and also discussion, to this direction would make this paper more interesting among the wider audience.

Review form: Reviewer 2

Recommendation

Accept with minor revision (please list in comments)

Scientific importance: Is the manuscript an original and important contribution to its field?

Excellent

General interest: Is the paper of sufficient general interest?

Excellent

Quality of the paper: Is the overall quality of the paper suitable?

Excellent

Is the length of the paper justified?

Yes

Should the paper be seen by a specialist statistical reviewer?

No

Do you have any concerns about statistical analyses in this paper? If so, please specify them explicitly in your report.

No

It is a condition of publication that authors make their supporting data, code and materials available - either as supplementary material or hosted in an external repository. Please rate, if applicable, the supporting data on the following criteria.

Is it accessible?

Yes

Is it clear?

Yes

Is it adequate?

Yes

Do you have any ethical concerns with this paper?

No

Comments to the Author

The paper is original, approaches an important group of Sphingidae, has well-structured phylogenetic results and a well-defined algorithm for biogeographical analysis. I am not particular fond of the "event-based likelihood ancestral area estimation", but it is a valid, transparent choice for reconstruction of the biogeographical history of a group – and generates a hypothesis to be tested.

My criticisms to the paper are two-folded, one on the assumption of origin of the group, one on the geological scenario where the evolution of the group took place.

On the question of origin of the group, on p. 3, the authors state "_Xylophanes_ belongs to the mostly Old World subtribe Choerocampina [19,20], suggesting that _Xylophanes_ may have dispersed to the New World via a jump dispersal event [19]." I understand that this is not the core of the paper, but a mistaken assumption about the origin of the clade may induce a mistaken perspective of the process as a whole. There was no previous solution to the evolution of transtropical elements in the Neotropical region than a "jump". The solution, however, is naïve and there is now a suitable alternative. Amorim et al. (2018) [Amorim, D.S., Oliveira, S.S. & Henao-Sepúlveda, A.C. 2018. A new species of _Eumanota_ Edwards (Diptera: Mycetophilidae: Manotine) in Colombia: a pseudogondwanan pattern. *American Museum Novitates*, 3915: 1-19.] recalls the presence of a wide tropical environment over Laurasian terranes (including the Oriental region) along the first half of the Cenozoic, drastically wiped out in large portions of Europe and North America along the second half of the Cenozoic with a sequence of glaciations. This is well documented and well known for vertebrates, although not towards the reconstruction of a Laurasian tropical continuum that is split by vicariance. There is no reason to assume any jump. Probably the Choerocampina were already there in the northern Neotropics/southern Nearctics before the Miocene and there is no need to advocate the clade "moving" into the Neotropical region from the Old World.

The second issue is about the geological context of the biogeographical interpretation of the evolution of the genus. The Discussion section is too brief and there should be at least a synthesis of what was going on since the Miocene from a geological perspective. What was the position of the Caribbean in relation to Central America in the early Miocene? How far north were the Andes raised in the early Miocene and along the subsequent ages? What about the big lake in the Amazon in the Miocene?

Even without a detailed association between event and cause of the event, the authors should bring in the general geological context in which occurred the assumed biogeographical evolution of the group.

Decision letter (RSPB-2021-2435.R0)

22-Dec-2021

Dear Dr Li

I am pleased to inform you that your manuscript RSPB-2021-2435 entitled "A diversification relay race from Caribbean-Mesoamerica to the Andes: Historical biogeography of *Xylophanes*, the most species rich hawkmoth genus" has been accepted for publication in Proceedings B.

The referee(s) have recommended publication, but also suggest some minor revisions to your manuscript. Therefore, I invite you to respond to the referee(s)' comments and revise your manuscript. Because the schedule for publication is very tight, it is a condition of publication that you submit the revised version of your manuscript within 7 days. If you do not think you will be able to meet this date please let us know.

Online supplementary material will also carry the title and description provided during submission, so please ensure these are accurate and informative. Note that the Royal Society will not edit or typeset supplementary material and it will be hosted as provided. Please ensure that

the supplementary material includes the paper details (authors, title, journal name, article DOI). Your article DOI will be 10.1098/rspb.[paper ID in form xxxx.xxxx e.g. 10.1098/rspb.2016.0049].

It is a condition of publication that data supporting your paper are made available either in the electronic supplementary material or through an appropriate repository. Please see our Data Sharing Policies <https://royalsociety.org/journals/authors/author-guidelines/#data>.

Sincerely,

Dr Maurine Neiman

Associate Editor

Comments to Author:

This MS is now been reviewed by two reviewers and both have been positive. One of the suggestions is that the introduction needs to highlight the importance of such taxa specific studies in understanding diversification in Neotropical and Mesoamerican areas. I hope detailed comments by reviewers will be helpful in revising the MS. Also, authors could consider removing "the most species rich hawmoth genus", for the sake of brevity of the title.

Reviewer(s)' Comments to Author:

Referee: 1

Comments to the Author(s)

I like this paper and recommend its publication in Proceedings B. Both the genomic and taxonomic sampling are comprehensive and enable assessing the addressed questions well. I also

did not detect any flaws with the conducted analyses although I am not familiar with all of them. The pictures provided are nice and informative. My only comment is that I find that the paper would benefit for putting the problem into a wider context. The scope of this paper is a bit narrow presently. Why is it important to understand phylogeography of these hawkmoths? Certainly this is of interest because it provides important insights on the species diversification more generally in the Neotropical and Mesoamerican areas. I think that the big question of biogeography is that why is there disproportionately high number of species in the Neotropical area compared to other areas, including tropical Asia and Africa. I find that expanding the introduction, and also discussion, to this direction would make this paper more interesting among the wider audience.

Referee: 2

Comments to the Author(s)

The paper is original, approaches an important group of Sphingidae, has well-structured phylogenetic results and a well-defined algorithm for biogeographical analysis. I am not particular fond of the "event-based likelihood ancestral area estimation", but it is a valid, transparent choice for reconstruction of the biogeographical history of a group – and generates a hypothesis to be tested.

My criticisms to the paper are two-folded, one on the assumption of origin of the group, one on the geological scenario where the evolution of the group took place.

On the question of origin of the group, on p. 3, the authors state "Xylophanes belongs to the mostly Old World subtribe Choerocampina [19,20], suggesting that Xylophanes may have dispersed to the New World via a jump dispersal event [19]." I understand that this is not the core of the paper, but a mistaken assumption about the origin of the clade may induce a mistaken perspective of the process as a whole. There was no previous solution to the evolution of trans-tropical elements in the Neotropical region than a "jump". The solution, however, is naïve and there is now a suitable alternative. Amorim et al. (2018) [Amorim, D.S., Oliveira, S.S. & Henao-Sepúlveda, A.C. 2018. A new species of Eumanota Edwards (Diptera: Mycetophilidae: Manotine) in Colombia: a pseudogondwanan pattern. *American Museum Novitates*, 3915: 1-19.] recalls the presence of a wide tropical environment over Laurasian terranes (including the Oriental region) along the first half of the Cenozoic, drastically wiped out in large portions of Europe and North America along the second half of the Cenozoic with a sequence of glaciations. This is well documented and well known for vertebrates, although not towards the reconstruction of a Laurasian tropical continuum that is split by vicariance. There is no reason to assume any jump. Probably the Choerocampina were already there in the northern Neotropics/southern Nearctics before the Miocene and there is no need to advocate the clade "moving" into the Neotropical region from the Old World.

The second issue is about the geological context of the biogeographical interpretation of the evolution of the genus. The Discussion section is too brief and there should be at least a synthesis of what was going on since the Miocene from a geological perspective. What was the position of the Caribbean in relation to Central America in the early Miocene? How far north were the Andes raised in the early Miocene and along the subsequent ages? What about the big lake in the Amazon in the Miocene?

Even without a detailed association between event and cause of the event, the authors should bring in the general geological context in which occurred the assumed biogeographical evolution of the group.

Author's Response to Decision Letter for (RSPB-2021-2435.R0)

See Appendix A.

Decision letter (RSPB-2021-2435.R1)

10-Jan-2022

Dear Dr Li

I am pleased to inform you that your manuscript entitled "A diversification relay race from Caribbean-Mesoamerica to the Andes: Historical biogeography of *Xylophanes* hawkmoth" has been accepted for publication in Proceedings B.

If you are likely to be away from e-mail contact please let us know. Due to rapid publication and an extremely tight schedule, if comments are not received, we may publish the paper as it stands. If you have any queries regarding the production of your final article or the publication date please contact procb_proofs@royalsociety.org

Data Accessibility section

Open Access

Paper charges

Sincerely,

Proceedings B

Appendix A

Dear Editor,

Thank you very much for organizing reviews and providing comments on our paper.

We have modified our title based on the Editor's comments.

According to referee 1's comments, we have modified the Introduction and Discussion parts to broaden the scope of our paper.

According to referee 2's comments, we have modified the Introduction part and added the alternative scheme of the origin of the clade. We have added a synthesis in the Discussion part about the changes in the Panama Isthmus, Andes, and Acre system since the Miocene, and their potential impact on the biogeographical evolution of the group.

Detailed modifications see tracked changes below.